# High prevalence of Panton-Valentine leukocidin positive, multidrug resistant, Methicillin-resistant *Staphylococcus aureus* strains circulating among clinical setups in Adamawa and Far North regions of Cameroon

**Mansour Mohamadou**[1,2,3], **Sarah Riwom Essama**[2], **Marie Chantal Ngonde Essome**[3], **Lillian Akwah**[2,3], **Nudrat Nadeem**[1], **Hortense Gonsu Kamga**[4], **Sadia Sattar**[1], **Sundus Javed**[1] *

**1** Biosciences Department, COMSATS University Islamabad, Islamabad, Pakistan, **2** Department of Microbiology, Faculty of Science, University of Yaounde 1, Yaoundé, Cameroon, **3** Centre for Research on Health and Priority Pathologies, Institute of Medical Research and Medicinal Plants Studies, Yaoundé, Cameroon, **4** Department of Microbiology, Haematology and Infectious diseases, Faculty of Medicine and Biomedical Sciences of University of Yaounde 1, Yaoundé, Cameroon

* sundus.javed@comsats.edu.pk

## Abstract

*Staphylococcus aureus (S. aureus)* is one of the earliest pathogens involved in human infections, responsible for a large variety of pathologies. Methicillin was the first antibiotic used to treat infections due to *S. aureus* but infections due to Methicillin resistant *Staphylococcus aureus* (MRSA) originated from hospital settings. Later, severe infections due to MRSA without any contact with the hospital environment or health care workers arose. Prevalence of MRSA has shown an alarming increase worldover including Cameroon. This Cross-sectional study was designed to evaluate the occurrence of MRSA infections in five different, most frequented Hospitals in northern Cameroon. Socio demographic data was recorded through questionnaire and different clinical specimens were collected for bacterial isolation. Identification of *S. aureus* was confirmed via 16s rRNA amplification using *S. aureus* specific primers. Molecular characterisation was performed through *mecA* gene, *Luk PV* gene screening and SCCmec typing. A total of 380 *S. aureus* clinical isolates were obtained of which 202 (53.2%) were nonduplicate multidrug resistant isolates containing, 45.5% MRSA. Higher number of MRSA was isolated from pus (30.4%) followed by blood culture (18.5%), and urine (17.4%). Patients aged 15 to 30 years presented high prevalence of MRSA (30.4%). Majority isolates (97.8%) carried the m*ecA* gene, PVL toxin screening indicated 53.3% isolates carried the *lukPV* gene. Based on PVL detection and clinical history, CA-MRSA represented 53.3% of isolates. SCCmec typing showed that the Type IV was most prevalent (29.3%), followed by type I (23.9%). Amongst MRSA isolates high resistance to penicillin (91.1%), cotrimoxazole (86.7%), tetracycline (72.2%), and ofloxacin (70.0%) was detected. Meanwhile, rifampicin, fusidic acid, lincomycin and minocycline presented high efficacy in bacterial control. This study revealed a high prevalence of MRSA among

**Data Availability Statement:** All relevant data are within the paper and its Supporting Information files.

**Funding:** Mohamadou Mansour was financially supported during his stay in Pakistan by Comsats University Islamabad and The World Academy of Sciences (TWAS) sandwich postgraduate fellowship award (FR number : 3240315420). No : The funders had no role in study design, collection and interpretation of data, decision to publish, no fund was received for publication. There was no additional external funding received for this study and we have not received any funds to cover APC fees.

**Competing interests:** The authors have declared no competing interest exist

infections due to *S. aureus* in Northern Cameroon. All MRSA recorded were multidrug resistant and the prevalence of CA MRSA are subsequently increasing, among population.

## Introduction

*Staphylococcus aureus (S. aureus)* is recognized as one of the earliest pathogens involved in human infections and was first time isolated from pus by Louis Pasteur [1]. This pathogen is responsible for a large variety of pathologies worldwide and is still one of the most common infections in humans affecting all social groups [2, 3]. Infections caused by this bacteria include skin and soft tissue infections, bacteremia, endocarditis, central nervous system infections, pulmonary infections, muscle and skeletal infections, genitourinary tract infections, as well as toxin-induced diseases such as gastroenteritis [4–6].

Methicillin was the first antibiotic used to treat infections due to *S. aureus* but in 1961 resistant strains were discovered in the UK [7, 8]. Since this discovery, mortality, and morbidity of patients suffering from Stapholoccocal infection is increasing worldwide [9, 10]. Infections related to Methicillin resistant *Staphylococcus aureus* (MRSA) include pyogenic endocarditis, suppurative pneumonia, osteomyelitis and pyogenic infections of the skin, soft tissues [11]. MRSA originated from hospital settings, termed Hospital Acquired MRSA (HA-MRSA) [9] but since 1990, patients, particularly the young started to develop infections due to MRSA without any contact with the hospital environment or health care workers, and strains isolated were hence termed Community-Acquired MRSA (CA-MRSA) [9, 12]. This advent of CA-MRSA has drastically changed the epidemiology of past century MRSA isolates [13]. MRSA strains were favored by the carriage of certain genes such as *mecA*, which has a genetic element called staphylococcal chromosome cassette (SCCmec) [14, 15]. SCCmec consists of two essential elements, the mec complex, consisting of *mecA* and its regulators, and the cassette chromosome recombinase (ccr) genes that ensure the mobility of the cassette important for bacterial survival in stress conditions [16, 17]. SCCmec also encodes including a cytolysin *psm-mec* [17] which has an important role in virulence. Moreover, the comparison of *mecA* homologues and their neighboring genes carried by SCCmec are useful in determining phylogeny of the species in an evolutionary perspective [18]. Many SCCmec sequence types have been registered [17]. For genotypic differentiation of CA and HA MRSA, data indicate that CA-MRSA belong to SCCmec types IV and V [12]. In addition, the severity of *S. aureus* infection is related to its ability to adapt to human immune system through the production of diverse virulence factors that counteract the innate immune response and delay the adaptive immune response, promoting bacterial spread to deep tissues and organs. One such virulence factor that mainly targets leukocytes, is the pore forming Panton Valentine Leukocidin (PVL) [18, 19]. PVL toxin has two components called *LukS-PV* and *LukF-PV* and exhibits both toxic and immunomodulatory properties and is associated with severe infection outcomes including necrotizing pneumonia, pyomyositis, brain abscess, necrosis, and apoptosis [20–22]. CA-MRSA carry SCCmec types IV and V, and secrete the PVL toxin [12, 13]. While the HA MRSA strains which cause nosocomial infections carry SCCmec types I, II, and III [23, 24]. These strains do not secrete PVL toxin in general [25]. In addition, HA MRSA is resistant to non-Beta lactam antibiotics including aminoglycopeptides, fluoroquinolones, and macrolides [26, 27], while CA MRSA demonstrate resistance to Beta lactams, fluoroquinolones, and tetracyclines [28–30]. In Cameroon, the differentiation of these two types of strains is set up following the CDC definition based on surgical intervention, hospital environment contact, and hospital stay [31].

Multidrug resistance has became a major public health concern all over the world. Recently, many authors have reported the emergence of multidrug resistant bacterial pathogens, from various origins including veterinary and humans health systems [31–33]. Prevalence of MRSA shows an alarming increase in Cameroon with an increase from 20–30% in 2003 [34], to 34.6% in 2013 [35], and 78.6% from 2017 to 2019 [36]. The aim of this study was to evaluate distribution of CA and HA MRSA in Cameroon and assess their genetic diversity through *MecA* gene, *Luk* PV screening as well as SCCmec typing. Furthermore, antimicrobial resistance profiling is performed as a step towards better regulation of MRSA strains in circulation among hospitals and communities of Cameroon in sub-Saharan Africa.

## Material and methods

A Cross-sectional study was conducted from April 2019 to December 2020 in five different, most frequented hospitals of Adamaoua and Far North Regions of Cameroon. Approval to conduct the study was obtained from the National Ethics Committee of Cameroon (N$^o$2017/12/958/CE/CNERSH/SP). Written authorization from each regional delegate of health and the various directors of hospitals were also obtained. Data were collected by questionnaire after the oral and written consent of patients or guardians. A literature review was conducted to design the questionnaire [37, 38]. Data acquired through questionnaire included sociodemographic information, Out or in- patients (according to the clinical definition of out or in patient care related to the CDC reference [31].

### Bacterial isolation and identification

Bacterial trains were isolated from eight different types of clinical samples including pus, urine, blood culture, surgery wound, vaginal swab, urethral swab, semen culture, stool culture. Mannitol salt Agar (Bio-Rad, France) was used for culture.

Preliminary identification was carried out using a combination of colony morphology observation, Gram staining, catalase test to differenciate between catalase positive staphylococcal vs catalase negative streptococcus isolates [39]. Furthermore, mannitol fermentation, coagulase test (to differentiate *Staphylococcus aureus* isolates from others species of *Staphylococcus*) [40], and Dnase test (for identification of *S. aureus*) were performed as described earlier [41]. Isolate details are included in S2 Table.

### Antimicrobial susceptibility

Antimicrobial susceptibility profiling was performed using the Kirby Bauer disc diffusion method on Muller Hinton Agar (Oxoid Ltd, Basingstoke, UK) according to the instruction of the European Committee on Antimicrobial Susceptibility Testing [42]. The turbidity of bacterial suspension was standardized by using 0.5 McFarland. A sterile cotton swab was dipped into the bacterial suspension and spread over the entire surface of Muller Hinton agar (Oxoid Ltd, Basingstoke, UK). Plates were left at room temperature for 3 to 5 min. Then, antimicrobial drug discs were placed by using a disc manual dispenser on to the Muller Hinton agar and incubated at 37˚C for 18–24 h. Sixteen antibiotics (Oxoid Ltd, Basingstoke, UK) from eight classes of antibiotics were tested, namely: Penicillin (Oxacillin (5μg), Amoxicillin and Clavulamic acid (30μg),); Cephalosporin (Cefoxitin (30μg)), Novobiocin (5μg), Fluoroquinolon (Ofloxacin (15μg), Ciprofloxacin (5μg),): Aminoglycoside (Gentanicin (10 μg),): Macrolide/Lincosamide (Pristinamicin (15μg), Erythromycin (15μg), Lincomycin (15μg)); Tetracycline (Tetracyclin (30μg) and Minocyclin (30μg)); Others (Cotrimoxazol (75μg), Rifampicin (30μg), Fusidic Acid (10μg)); and Glycopeptide (Vanconicin (30μg)). At the end of the incubation period, the diameter of the growth inhibition area was measured by a digital caliper. The

corresponding measured diameters were interpreted as susceptible (S), intermediate (I), or resistant (R) according to the European Committee on Antimicrobial Susceptibility Testing guidelines. Bacteria resistant to at least one antibiotic from three or more different antimicrobial classes were considered as MDR bacteria [43]. Methicillin resistance was characterized by Oxacillin (5μg) and Cefoxitin (30μg) double disc diffusion test. Isolates showing inhibition zone < 26 mm for oxacillin and < 22 mm for cefoxitin were considered methicillin-resistant *Staphylococcus aureus* (MRSA) according to the European Committee on Antimicrobial Susceptibility Testing and Antibiotic Committee of France Society of Microbiology 2019 [42].

## Molecular identification of MRSA

**DNA extraction.** Methicillin-resistant *Staphylococcus aureus* (MRSA) isolated in this study were inoculated using Mannitol Salt Agar (Oxoid Ltd, UK) and incubated at 37˚C for 24 hours. DNA extraction and purification of *S. aureus* was performed using the Solis Bio-Dyne DNA extraction kit (Solis BioDyne, Estonia) according to manufacturers' instructions [44, 45]. A couple of checks were realized to confirm DNA extraction: Evaluation of quality and quantity of DNA using Nanodrop machine and Gel electrophoresis using 1% agarose gel.

**PCR confirmation of *Staphylococcus aureus*.** Molecular identification of *S. aureus* strains was performed via amplification of the conserved 16s rRNA region in *S. aureus* amplifying a 409 kb product using 5'-ATTAGATACCCTGGTAGTCCACGCC- 3' and 5'-CGTCATCCC CACCTTCCTCC-3' primers [46]. A total volume of 20 μL of PCR reaction mixture was constituted by adding 09 μL of PCR water, 1μL of forward, and reverse primers each, 04 μL of Master Mix (5x) (Solis BioDyne) and 05 μL of template DNA. Following conditions were established for this PCR: Initial denaturation 94˚C for 5 min; denaturation 94˚C for 45 seconds; annealing 60˚C for 45 seconds; amplification 72˚C for 30 seconds for 30 cycles; Final extension was performed at 72˚C for 10 minutes. PCR products were resolved on 1% agarose gel at 90V for 40 min and visualized on UV transilluminator. Amplicon size of 409bp indicated the presence of *S. aureus*.

**Detection of *mecA* and *PVL*.** *mecA* and *PVL* were detected using a multiplex PCR screening method. *MecA* gene was amplified using mecA1 (5'-GTAGAAATGACTGAACGTCCG ATAA-3'') and mecA2 (5'-CCAATTCCACATTGTTTCGGTCTAA-3'') primers. Panton-Valentine Leucocidin was identified using primers sequence detecting *lukS/F-PV* genes. *S/F* primers use were *Luk PV1* (5'-ATCATTAGGTAAAATGTCTGGACATGATCCA-3'') and *Luk PV2* (5'-GCATCAASTGTATTGGATAGCAAAAGC-3*). The mixture was prepared by adding 1.5 μL of each of the four primers, 10 μL of PCR water, Master Mix 4 μL (Solis bioDyne), and template DNA 05 μL. While the thermocycling conditions were: Initial denaturation 95˚C for 5 min; denaturation 95˚C for 1 min; annealing 55˚C for 1 min; amplification 72˚C for 1 min 30 seconds; Final extension 72˚C for 10 minutes for 30 cycles then resting at 4˚C [38]. PCR products were resolved on 1% agarose gel at 90V. The gels were visualized under UV light to detect the expected band sizes (433 bp for *Luk S/F* and 310 for mec A) [47, 48].

**Detection of Sccmec types I, II, III, IV, and V.** In general, Community-acquired MRSA (CA-MRSA) carry *PVL* encoding genes *LukS-PV* and *LukF-PV* which are associated with increased virulence and HA-MRSA lack this gene. However, recent evidence shows that this difference has gradually become indistinct in recent CA and HA MRSA strains [13, 49]. For clear stratification of HA-MRSA and Community CA-MRSA, identification of Sccmec (Staphylococcal Chromosome Cassette mec) types is useful. HA-MRSA strains carry SCCmec types I, II, or III and do not have *PVL* encoding genes [25], while CA-MRSA carry SCCmec IV and

V [50, 51]. The protocols used by Boyle et al and McClure-Warnier were followed for the molecular identification of Sccmec types I to V [44]. The reaction mixture was prepared as earlier but for the addition of oligomers mentioned in S1 Table.

Thermocycling parameters followed were: 95˚C for 4 min, followed by 34 cycles of 95˚C for 1 min, 53˚C for 1 min, 72˚C for 1 min 30 sec and final extension at 72˚C for 4 min at 4˚C. The PCR products were electrophoresed on 2% agarose gel containing ethidium bromide and visualized under UV transilluminator.

**Statistical analysis.**   Data were maintained in Excel sofware, and exported to SPSS version 25.0 where statistical analysis like percentages, proportion and association were performed. Graphs were prepared using originPro 9.0 64 bit sofware. Descriptive statistics like mean, frequencies were performed on different variables. Chi-square and Fisher exact test were used to test categorical variables. A significant difference was considered at a P-value < 0.05.

## Results

### Phenotypic characteristics of the recovered isolates

During the nineteen (19) months of collection, 630 samples of Gram positive cocci were recorded from five clinical laboratory hospitals in northern Cameroon. Based on colony morphology observation, Gram staining, catalase, mannitol fermentation, coagulase, and Dnase test, 380/630 (60.3%) samples were identified as *S. aureus*, among these 201 (52.9%) originated from Adamaoua region and 179 (47.1%) from far north. Results of antimicrobial susceptibility showed that 202/380 (53.2%) of *S.aureus* were resistant to at least three antibiotics (multi drug resistant) from three different classes. Subsequent analysis was focused on these 202 nonduplicate multidrug resistant samples of *Staphylococcus aureus*. High prevalence of *S. aureus* was identified from pus 25.7% followed by urine 20.8%, urethral swab 12.5% and blood culture 11.2%. Among these MDR strains, 92 (45.5%) were MRSA (Table 1).

Socio demographical data of study participants showed that men were most represented (68.5%) compared to women (31.5%). The age of the patients ranged from 21 days to 85 years. Patients aged 15 to 30 presented the highest prevalence of MRSA (30.4%), followed by the 0 to 15 years age group. Most samples belonged to the Adamawa region (56.5%) while 43.5% were isolated from the Far North region of Cameroon. A higher number of MRSA were isolated from pus (30.4%) followed by blood culture (18.5%), and urine (17.4%). Urethral and vaginal isolates were equally represented with 9.8% isolates from these biological samples (Table 2).

Based on the CDC definition of Community-Acquired Methicillin-Resistant *Staphylococcus aureus* (CA-MRSA) which relies on surgical intervention, hospital environment contact, and hospital stay (32), 56.5% of the clinical isolates were categorized as CA-MRSA versus 43.5% Hospital-Acquired MRSA (HA-MRSA) isolates. The difference between CA-MRSA and HA-MRSA based on gender was not statistically significant (p > 0.05), while a significant

**Table 1.  Prevalence of *Staphylococcus aureus* among clinical samples collected between April 2019 to December 2020.**

|  | Regions | | |
|---|---|---|---|
|  | **Adamawa (%)** | **Far north (%)** | **Total of samples** |
| Gram positive cocci | 332 (52.7) | 298 (47.7) | 630 |
| *S. aureus* | 201 (52.9) | 179 (47.1) | 380 |
| Multi drug resistant (MDR) *S. aureus* | 120 (59.4) | 82 (40.6) | 202 |
| Methicillin resistant *S. aureus* (MRSA) | 60 (57.7) | 44 (42.3) | 104 |
| Methicillin resistant *S. aureus* after molecular confirmation | 52 (56.5) | 40 (43.5) | 92 |

**Table 2. Repartition of MRSA based on age group and clinical sources.**

|  | HA-MRSA (%) | CA MRSA (%) |
|---|---|---|
| **Age group** |  |  |
| **0–15** | 18 (66.7) | 9 (33.3) |
| **15–30** | 4 (14.3) | 24 (85.7) |
| **30–45** | 7 (35) | 13 (65) |
| **45–60** | 2 (50) | 2 (50) |
| **60–75** | 6 (60) | 4 (40) |
| **≥75** | 3 (100) | 0 (00) |
| **Clinical source** |  |  |
| **Urine** | 4 (25) | 12 (75) |
| **Pus** | 18 (64.3) | 10 (35.7) |
| **Vaginal swab** | 1 (11.1) | 8 (88.9) |
| **Blood culture** | 12 (70.6) | 5 (29.4) |
| **Stool culture** | 0 (00) | 4 (100) |
| **Semen culture** | 0 (00) | 4 (100) |
| **Uretral collection** | 0 (00) | 9 (100) |
| **Sugery wound** | 5 (100) | 0 (00) |

difference was observed between CA-MRSA, HA-MRSA concerning age group and clinical samples (p<0.05) (Table 2).

## Distribution of *LukS/F PV* genes and *SCCmec* typing

Multiplex PCR detection of *mecA* and *PVL* genes revealed that 97.8% (n = 90) of our samples carried the m*ecA* gene. While *PVL* toxin screening indicated that 53.3% (n = 49) carried the *lukPV* gene, 44.6% (n = 41) were *Luk S/F* negative. SCCmec typing showed that the Type IV was most prevalent (29.3%), followed by type I (23.9%), type V (22.8%), type III (14,1%), and type II (7.6%). Meanwhile, 2.2% isolates were not typable. Classification of Hospital Acquired MRSA (HA-MRSA) and Community-Acquired MRSA based on *PVL* production and SCCmec types showed that the CA MRSA (SCCmec types IV and V producing *PVL* toxin) was most prevalent 52.1% (n = 48) while 45.6% (n = 42) were HA MRSA (SCCmec types I, II and III and non producers of *PVL*). Similar distribution (CA-MRSA 56.5% and HA-MRSA 43.5%) was observed following the CDC definition as mentioned before, and there is no statistical difference between both methods with p = 0.577.

The most prevalent SCCmec type in all clinical samples was SCCmec Type IV (n = 27) and type I (n = 22). SCCmec type I was most prevalent in pus (n = 12) samples followed by type V (n = 7) which was most commonly observed in urine, and blood culture (n = 5). SCCmec type II was the least common and found in only 3 different isolates. Meanwhile type III was present in all the clinical samples, from pus (23.1%), urine (15.4%), and stool (7.7%) sources. Similarly, SCCmec type IV was common in isolates from pus (25.9%) and blood (33.5%) (Table 3).

The geographical distribution of data showed that patients from Adamaoua region were most represented 52 (56.5%) than those from the far north region 40 (43.5%). While CA and HA MRSA repartition showed that HA MRSA were the most prevalent 21 (52.5%) in the Far north meanwhile CA MRSA were mostly present from out patient in the Adamaoua region 57.7% (n = 30) (Table 2). For SCCmec distribution, the type IV and V (n = 15 for each) were mostly represented in the Adamawa region followed by type I and III (n = 9 for each) while in the far north region the type I (n = 14) was most prevalent followed by type IV (n = 13) (Table 4).

**Table 3. SCCmec typing of MRSA isolates.**

| | | Urine (%) | Pus (%) | Vaginal cult (%) | Blood cult (%) | Stool (%) | Semen (%) | Urethral (%) | Surgery wound(%) | Total |
|---|---|---|---|---|---|---|---|---|---|---|
| *SCCMec* types | I | 00 (00) | 12(54,6) | 2(9.1) | 2(9.1) | 2(9.1) | 1(4.5) | 2(9.1) | 1(4.5) | 22 |
| | II | 2(28.7) | 1(14.2) | 0(00) | 2(28.7) | 1(14.2) | 0(00) | 1(14.2) | 0(00) | 7 |
| | III | 02(15.4) | 3(23) | 2(15.4) | 2(15.4) | 1(7.7) | 1(7.7) | 1(7.7) | 1(7.7) | 13 |
| | IV | 03(11.1) | 7(26) | 2(7.4) | 9(33.3) | 0(00) | 1(3.7) | 3(11.1) | 2(7.4) | 27 |
| | V | 07(33.3) | 5(23.8) | 3(14.3) | 2(9.5) | 0(00) | 1(4.8) | 2(9.5) | 1(4.8) | 21 |
| | Not typable | 2(100) | - | - | - | - | - | - | - | 2 |
| Total | | 16 | 25 | 9 | 15 | 4 | 4 | 8 | 5 | 92 |

## Antimicrobial susceptibility testing of isolates

All clinical *S. aureus* isolates were processed for antimicrobial susceptibility and classified as Multi drug-resistance (MDR), Extensively drug-resistance (XDR) and pandrug resistance (PDR) based on definition given by Magiorakos *et al.* [43].

MRSA isolates presented high resistance to penicillin (91.1%) followed by cotrimoxazol (86.7%), tetracycline (72.2%), and ofloxacin (70.0%). All (100%) MRSA isolates were multi-drug resistant. While rifampicin, fusidic acid, and minocycline presented high susceptibility respectively 90.0%, 75.6%, and 64.4% (Table 5).

Difference between, CA and HA MRSA is made on the basis of SCCmec and *PVL* secretion, antimicrobials comparison of these two types showed that the CA-MRSA were mostly resistant to beta-lactams, fluoroquinolones, tetracyclines, such as penicillin (95.6%), cotrimoxazole (87.4%), ofloxacin (79.2%) and tetracycline (75.0), meanwhile, HA-MRSA were most resistant to penicillin (85.7%), cotrimoxazole (85.7%), tetracycline (69.0%), ofloxacin (59.5%), gentamicin (47.6%). While most isolates were susceptible for rifampicin (88.1%), fusidic acid (76.2%), and lincomycin (66.7%) (Fig 1). HA MRSA are known to be resistant to non-Beta lactam antibiotics including aminoglycopeptides, fluoroquinolones, and macrolides [25] while CA MRSA demonstrate resistance to Beta lactams, fluoroquinolones, and tetracyclines [27]. Our study shows extensive drug resistance among CA MRSA that is steadily increasing in prevalence compared to previous data. It is alarming to note that while all MRSA were MDR, certain PDR isolates among both HA MRSA and CA MRSA were also identified (Table 6).

## Distribution of SCCmec types according to antibiotic resistance phenotypes for MRSA

Distribution of SCCmec types according to antibiotic resistance phenotypes for MRSA isolates shows that sccmec type IV and V, which cause infections in community, were mostly resistant to penicillin, ofloxacine, cotrimoxazole and tetracycline. Meanwhile type I, II and III,

**Table 4. Region wise distribution of SCCmec types among clinical samples collected from the Adamawa and Far north regions of Cameroon.**

| Types of Sccmec | Adamawa region (%) | Far north region (%) |
|---|---|---|
| I | 9 (40.9) | 13 (59.1) |
| II | 3 (42.9) | 4 (57.1) |
| III | 9 (69.2) | 4 (30.8) |
| IV | 15 (55.6) | 12 (44.4) |
| V | 15 (71.4) | 6 (28.6) |
| Not typable | 2 (100) | 0 |

**Table 5. Antimicrobial susceptibility profile of MRSA isolates.**

| Group of ATB | ATB tested | Antibiotic susceptibility profile | | |
|---|---|---|---|---|
| | | R (%) | S (%) | I (%) |
| Penicillin | AMC | 43 (47.8) | 11 (12.2) | 36 (40.0) |
| | OX | 92 (100) | 00 (00) | 00(00) |
| | P | 82 (91.1) | 05 (05.6) | 03 (03.3) |
| Cephalosporin | FOX | 92 (100) | 00 (00) | 00 (00) |
| Fluouroquinolon | CIP | 39 (43.3) | 34 (37.8) | 17 (18,9) |
| | OFX | 63 (70.0) | 21 (23,3) | 06 (06,7) |
| Aminoside | GEN | 51 (56.7) | 28 (31.1) | 11 (12,2) |
| Macrolide /Lincosamide | E | 40 (44.4) | 43 (47.8) | 07 (7.6) |
| | L | 17 (18.9) | 58 (64.4) | 15 (16.7) |
| | PI | 03 (3.3) | 35 (38.9) | 52 (57.8) |
| Tetracycline | TET | 65 (72.2) | 21 (23.3) | 04 (04.4) |
| | MIN | 08 (08.9) | 54 (60.0) | 28 (31.1) |
| Others | SXT | 78 (86.7) | 08 (08.9) | 04 (04.4) |
| | RD | 07 (07.8) | 81 (90.0) | 02 (02.2) |
| | FA | 07 (07.8) | 68 (75.6) | 15 (16.7) |
| Glycopeptide | VA | 12 (13.3) | 34 (37.8) | 44 (48.9) |

responsible for hospital acquire infections were resistant to penicillin, cotromoxazole, tetracycline and erythromycine (Table 7).

## Correlation of clinical sources, and antimicrobial resistance

*S. aureus* isolated from all clinical samples were multi drug resistant. Samples from urine, blood culture and surgery wound were comming mostly from community sources, while *S.*

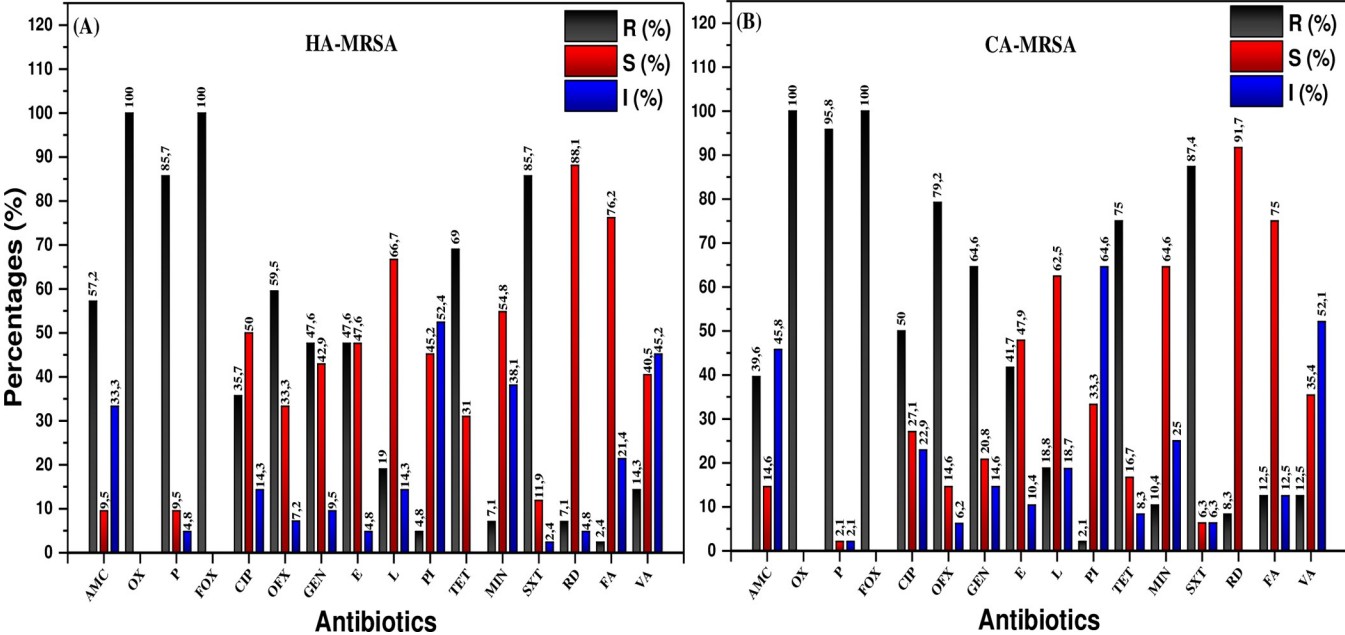

**Fig 1. Antimicrobial susceptibility profile of CA and HA MRSA.** Kirby-bauer disc diffusion method was used to test Antimicrobial susceptibility of CA and HA MRSA against Amoxicilline+Clavulanic acid (AMC), Oxacillin (OX), Cefoxitin (FOX), Ciprofloxacin (CIP), Ofloxacin (OFX), Gentamicin (GEN), Erythromycin (E), Lincomycin (L), Tétrecyclin (TET) Cotrimoxazole (SXT), Rifampicin (R), FusidicD Acid (FA), Vancomycin (VA), Penicillin (P), Minociclin (MI). Zones of inhibition were interpreted according to EUCAST guidelines, R = Resistance; S = Sensible; I = Intermediaire.

**Table 6. Antimicrobial susceptibility profile of *S. aureus*.**

| | Resistance profile | Most common antibiotic resistance patterns |
|---|---|---|
| Hospital acquired–MRSA | 6 MDR | Aminoside (GEN); Fluoroquinolone (OFX, CIP); Macrolides (E, L) |
| | 28 XDR | |
| | 2 PDR | |
| Community acquired-MRSA | 29 MDR | Betalactamines (P, OX, FOX, VA), Fluoroquinolone (CIP, OFX), Tetracycline (TET, MIN) |
| | 41 XDR | |
| | 2 PDR | |

*aureus* isolated from pus, stool culture were predominantly nosocomial. Our results showed that samples from CA MRSA, are commonly resistant to penicillin, cotrimoxazole, tetracycline and ofloxacine while rifampicin, lincomycin, erythromycin and minocycline presented high sensitivity rates. On the otherhand HA MRSA, were commonly resistant to penicillin, cotrimoxazole, tetracycline and gentamicine while showing sensitivity to rifampicin, lincomycin and fusidic acid (Table 8).

## Discussion

*S. aureus* is responsible for a large variety of pathologies worldwide [2].This pathogen has developed several resistance mechanismsagainst antimicrobial treatment. They include enzymatic inactivation of the antibiotic, alteration of the target with decreased affinity for the antibiotic, and also spontaneous mutations [44]. Methicillin-resistant *Staphylococcus aureus* remains an important issue associated with high mortality and morbidity of patients. MRSA are still a major public health concern worldwide, due to treatment challenges [52]. In the last century, infections related to MRSA only originated from hospitals known as Hospital-

**Table 7. Distribution of SCCmec types according to antibiotic resistance phenotypes for MRSA.**

| Antibiotic classes | Antibiotic Tested | Types of SCCmec | | | | |
|---|---|---|---|---|---|---|
| | | HA MRSA | | | CA MRSA | |
| | | I | II | III | IV | V |
| | | R (%) | R (%) | R (%) | R (%) | R (%) |
| Penicillin | Amoxicillin | 13 (59.1) | 3 (42.8) | 8 (61.5) | 8 (29.6) | 11(52.4) |
| | Oxacillin | 22 (100) | 7 (100) | 13 (100) | 27 (100) | 21(100) |
| | Penicillin | 20 (90.9) | 6 (85.7) | 10 (76.9) | 26 (96.3) | 20 (95.2) |
| Cephalosporin | Cefoxitin | 22 (100) | 7 (100) | 13 (100) | 27 (100) | 21 (100) |
| Fluoroquinolone | Ciprofloxacin | 7 (31.8) | 3 (42.3) | 5 (38.5) | 11 (40.7) | 13 (61.9) |
| | Ofloxacin | 13 (59.1) | 3 (42.3) | 9 (69.3) | 20 (74.1) | 18 (85.7) |
| Aminoside | Gentamicin | 9 (40.9) | 4 (57.1) | 7 (53.8) | 19 (70.4) | 12 (57.1) |
| Macrolide /Lincosamide | Erythromycin | 10 (45.5) | 3 (42.9) | 7 (53.8) | 10 (37.4) | 10 (47.6) |
| | Lincomycin | 4 (18.2) | 2 (28.6) | 2 (15.4) | 6 (22.2) | 3 (14.3) |
| | Pristynamicin | 2 (8.9) | 0 (0) | 0 (0) | 0 (0) | 1 (4.8) |
| Tetracyclin | Tetracycline | 13 (59.1) | 5 (71.4) | 11 (84.6) | 22 (81.5) | 14 (66.7) |
| | Minocyclin | 1 (4.5) | 1 (14.3) | 1 (7.7) | 4 (14.8) | 1 (4.8) |
| Others | Cotrimoxazol | 18 (81.8) | 7 (100) | 11 (84.6) | 24 (88.9) | 18 (85.7) |
| | Rifampicin | 1 (4.5) | 1 (14.3) | 1 (7.7) | 2 (7.4) | 2 (9.1) |
| | Fusidic acid | 0 (0) | 0 (0) | 1 (7.7) | 3 (11.1) | 3 (14.3) |
| Glycopeptide | Vancomycin | 4 (18.2) | 2 (28.6) | 0 (0) | 3 (11.1) | 3 (14.3) |

**Table 8. Correlation of clinical sources, and antimicrobial resistance profiles.**

| Clinical sample source | Sccmec Type | Common resistant antibiotics | Common susceptible antibiotics |
|---|---|---|---|
| Urine | 10/14 (71.4%) type IV to V | P, SXT, TET, OX, FOX | GEN, E, L, MI, RD |
| Pus | 16/28 (57.1%) type I to III | P, SXT, TET, AMX, OX, FOX, GEN | L, MI, RD, FA |
| Vaginal collection | 5/9 (55.6%) type IV to V | P, SXT, TET | GEN, E, L, MI, RD |
| Blood culture | 11/17 (64.7%) type IV to V | P, SXT, TET, OFX, GEN, | E, L, PI, MI, RD, FA |
| Stool culture | 4/4 (100%) type I to III | P, SXT, TET, GEN | RD |
| Semen culture | 2/4 (50%) type IV to V | P, SXT, TET | L, PI, RD |
| Urethral collection | 5/9 (55.6%) type IV to V | P, SXT, TET, OFX, | GEN, E, L, PI, RD, FA |
| Sugery wound | 3/5 (60%) type IV to V | P, SXT, TET, OFX, L | RD |

Amoxicilline+Clavulanic acid (AMC), Oxacillin (OX), Cefoxitin (FOX), Ciprofloxacin (CIP), Ofloxacin (OFX), Gentamicin (GEN), Erythromycin (E), Lincomycin (L), Tétrecyclin (TET) Cotrimoxazole (SXT), Rifampicin (RD), Fusidic Acid (FA), Vancomycin (VA), Penicillin (P), Minociclin (MI). MDR: Multi drugs resistance

acquired *Staphylococcus aureus* [9]. However since 1990, infections due to MRSA without any contact with hospital or health care workers arose, thereafter termed Community-Acquired *Staphylococcus aureus* (CA-MRSA) [12]. In Cameroon Prevalence of MRSA has been increasing steadily with a rise from 20–30% prevalence since 2003 [34] to 80% in 2019 [53]. In this study, 202 nonduplicate multi drug resistant samples of *S. aureus* were isolated from the five most frequented hospitals of Adamawa and Far North regions of Cameroon. Among these clinical isolates 45.5% were identified as MRSA. Repartition of our clinical samples indicates that MRSA were most frequently detected in pus followed by blood culture and urine with respectively 30.4%, 18.5%, and 17.4% prevalence. Pus samples were mostly obtained from hospitalised patients while urine and blood culture were from outpatient wards. It is important to indicate that we have mostly focused our attention on the urinary tract infections associated *S. aureus* isolated from pregnant women, young children aged less than 5 years, and people living with comorbidities such as HIV and diabetes because they have compromised or immature immune systems. Antimicrobial susceptibility testing showed high resistance to penicillin, cotrimoxazole, tetracycline, ofloxacine and gentamicine. A similar report from Nepal also showed inefficacy of these antibiotics for treatment of local isolates [54]. Meanwhile rifampicin, fusidic acid, lincomycin and minocycline showed efficacy against our isolates. Similar efficacy of rifampicin was also observed in isolates from Creech [55]. Sociodemographic distribution showed that men were most susceptible with 68.5% (n = 63) prevalence. Similar prevalence among men was also observed in Ethiopia [56] and Barbados [57]. Samples from urine, blood culture and sugery wound were acquired mostly from community sources and showed resistance to penicillin, cotrimoxazole, tetracycline and ofloxacine while rifampicine, lincomycin, erythromycin and minocycline presented high sensitity rate. Meanwhile *S. aureus* isolated from pus, stool culture were from inpatient source. These isolates were resistant to penicillin, cotrimoxazole, tetracycline and gentamicine, while sensitivity to rifampicine, lincomycin and fusidic acid were observed. Similar results were reported in India recently [12]. In Cameroon betalactams, cephalosporin, tetracycline are mostly used to treat infections related to *S. aureus* [35, 58]. However, emergence of multiresistant strains carrying the PVL toxin worsen the situation. Therefore rifampicine, lincomycin and minocycline are recommended for effective control. Our study also highlights the fact that the rate of multidrug resistant MRSA among population is increasing in Cameroon as all our MRSA isolates were MDR. Mainstay antimicrobials including beta-lactams, and aminoglycoside tetracyclin, frequently use in treatment presented high resistance rate amongst our isolates SCCmec typing, depicted that the SCCmec type IV (29.3%) was most dominant followed by type I (23.9%), type V

(22.8%) and type III (13.0%). This distribution was also observed in the study conducted in Uganda [52, 59]. On the basis of SCCmec molecular typing and *PVL* secretion, our study revealed that CA MRSA (which have SCCmec types IV and V) were more prevalent 52.1% than HA MRSA (Types I to III), as reported earlier among Dutch [60], American [61], and Arab populations [62]. The predominance of CA MRSA in our study can be explained by the fact that people aged 0 to 30 years were most represented (59.7%). Younger people mostly carry CA MRSA, due to infrequent hospitalization [63]. In addition, factors like poor hygiene, socio-cultural habits in these regions of study may contribute to increased prevalence of *S. aureus* as it is easily transmitted by hand contact. The increasing MDR MRSA strains in circulation in the youth is especially troubling as it comprises the majority of Cameroons population with major economic contribution. Moreover, our study describes a high prevalence of XDR and émergence of PDR among CA MRSA and HA MRSA. Alarming rate of XDR and PDR was also described by Haji *et al.* [64], Important meseares must be taken in Cameroon and over the world for limiting spread of these emerging resistant strains. Emergence of MDR MRSA harboring the PVL toxin worsen the situation in terms of infection control and therapy. Previously, the *lukSF-PV* genes were a frequent genetic marker of CA MRSA isolates with SCCmecIV or SCCmecV [65]. However, subsequent studies have demonstrated *lukSF-PV+* HA MRSA strains [66]. In this study both HA MRSA and CA MRSA strains harbored the *lukSF-PV* genes. CA MRSA infection and carriage of PVL toxin is thought to be responsible for rapidly progressive and lethal infections including necrotizing pneumonia, severe sepsis, and necrotizing fasciitis [67]. Therefore it has been speculated that *lukSF-PV+* HA MRSA might lead to severe clinical infections similar to that caused by *lukSF-PV+* CA MRSA [68]. This is consistent with the fact that the mechanism of PVL toxicity involves cell lysis in human myeloid cells, promoting inflammation and possesses immunomodulatory properties, blocking immune activation therefore promoting tissue damage [69]. Further studies need to be undertaken with indepth genotypic analysis of resistance mechanisms and correlation with phenotypic data which is a limitation in this study. Moreover, detailed virulence profiling of isolates may indicate better control measures.

## Conclusion

This study was designed to evaluate the prevalence of CA and HA MRSA circulating in major hospitals in Cameroon and assess their genetic diversity through *MecA* gene and PVL toxin screening. We observed a high prevalence of MDR, XDR and PDR MRSA strains as the etiological agent responsible for a wide variety of infections due to *Staphylococcus aureus*, in patients from Cameroon. While *lukSF-PV* genes were detected in both CA and HA MRSA, prevalence of PVL toxin secreting CA MRSA was alarmingly high, especially among patients below 30 years of age. Isolates depicted high resistance to betalactams, tetracyclin, fluouroquinolone meanwhile rifampicine, fusidic acid, lincomicine and minocycline presented high rate of sensitivity, and should subsequently be prescribed for *S. aureus* infection control. Rapid evolution of multidrugresistant CA MRSA strains observed in our society needs an action plan with a good diagnostic protocol before any treatment to reduce therapeutic failures. Also, implementation of strict aseptic measures for healthcare professionals and more education in hygiene measures are required to limit the spread of CA MRSA.

## Supporting information

**S1 Table. Oligonucleotide primers used for Sccmec types I to V identification [44].** (DOCX)

**S2 Table. Metadata for bacterial isolates from patients with MRSA infections.**
(XLSX)

**S1 File.**
(DOCX)

## Acknowledgments

We thank The World Academy of Sciences (TWAS) and COMSATS University Islamabad (CUI) for the sandwich postgraduate fellowship award offer to Mohamadou Mansour.

## Author Contributions

**Conceptualization:** Mansour Mohamadou, Sarah Riwom Essama, Hortense Gonsu Kamga, Sundus Javed.

**Data curation:** Mansour Mohamadou, Lillian Akwah, Nudrat Nadeem, Sadia Sattar.

**Formal analysis:** Mansour Mohamadou, Marie Chantal Ngonde Essome, Hortense Gonsu Kamga, Sundus Javed.

**Methodology:** Mansour Mohamadou, Marie Chantal Ngonde Essome, Nudrat Nadeem, Sadia Sattar, Sundus Javed.

**Project administration:** Sarah Riwom Essama, Hortense Gonsu Kamga, Sundus Javed.

**Resources:** Lillian Akwah, Sadia Sattar.

**Supervision:** Sarah Riwom Essama, Hortense Gonsu Kamga, Sundus Javed.

**Validation:** Sarah Riwom Essama, Sadia Sattar.

**Writing – original draft:** Mansour Mohamadou.

**Writing – review & editing:** Mansour Mohamadou, Marie Chantal Ngonde Essome, Lillian Akwah, Nudrat Nadeem, Sadia Sattar, Sundus Javed.

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
