## [Decision Letter · Decision Letter 0]

3 Jan 2022

PONE-D-21-37074Genotypic and antimicrobial profiling of Methicillin-resistant Staphylococcus aureus from Adamaoua and Far North Regions of CameroonPLOS ONE

Dear Dr. Javed,

Thank you for submitting your manuscript to PLOS ONE. After careful consideration, we feel that it has merit but does not fully meet PLOS ONE’s publication criteria as it currently stands. Therefore, we invite you to submit a revised version of the manuscript that addresses the points raised during the review process.

ACADEMIC EDITOR:Please revise the manuscript according to the reviewer comments. A major revision is required.

We look forward to receiving your revised manuscript.

Kind regards,

Abdelazeem Mohamed Algammal, Prof, Ph.D

Academic Editor

PLOS ONE

Journal Requirements:

https://journals.plos.org/plosone/s/file?id=ba62/PLOSOne_formatting_sample_title_authors_affiliations.pdf2. 

(Mohamadou Mansour was financially supported during his stay in Pakistan by Comsats University Islamabad and The World Academy of Sciences (TWAS) sandwich postgraduate fellowship award (FR number : 3240315420).

No : The funders had no role in study design, collection and interpretation of data, decision to publish, no fund was received for publication.)

Please include your amended Funding Statement within your cover letter. We will change the online submission form on your behalf."

Reviewers' comments:

Reviewer's Responses to Questions

**Comments to the Author**

1. Is the manuscript technically sound, and do the data support the conclusions?

Reviewer #1: Partly

Reviewer #2: Partly

2. Has the statistical analysis been performed appropriately and rigorously? 

Reviewer #1: N/A

Reviewer #2: Yes

3. Have the authors made all data underlying the findings in their manuscript fully available?

Reviewer #1: Yes

Reviewer #2: Yes

4. Is the manuscript presented in an intelligible fashion and written in standard English?

Reviewer #1: No

Reviewer #2: No

5. Review Comments to the Author

Reviewer #1: Comments to authors:

-The current study is interesting; however, the authors should address the following comments to improve the quality of the manuscript:

Title:

I think the work would benefit from the title that contains the main conclusion of the study (should be derived from the conclusion). Please modify the title.

Abstract:

- The abstract must illustrate the used methods and the most prevalent results (give more hints about methods and results). Besides, rephrase the aim of the work and the main conclusion of your findings.

Introduction: (it needs to be more informative)

-Give a hint about the virulence factors, different infections caused by MRSA, and the mechanism of disease occurrence.

- The authors should illustrate the public health importance concerning the emergence of multidrug-resistant (MDR) bacterial pathogens that reflect the necessity of new potent and safe antimicrobial agents. Several studies proved the widespread MDR- bacterial pathogens;

Authors could add the following paragraph:

Multidrug resistance has been increased all over the world that is considered a public health threat. Several recent investigations reported the emergence of multidrug-resistant bacterial pathogens from different origins including humans, birds, cattle, and fish that increase the need for routine application of the antimicrobial susceptibility testing to detect the antibiotic of choice as well as the screening of the emerging MDR strains. You should cite the following valuable studies:

1.PMID: 33177849

2. PMID: 32397408

3.PMID: 32994450

4. PMID: 32497922

5.PMID: 33061472

6.PMID: 33947875

7.PMID: 34445951

8.PMID: 33188216

9.https://doi.org/10.1016/j.aquaculture.2021.737643

10.PMID: 30150182

-Rephrase the aim of the work to be clear and better sound.

Material and methods: Illustrate your methods with subtitles:

-Add this subtitle: Bacterial Isolation and identification:

•Discuss in detail the methods of isolation and identification of S.aureus and MRSA. Besides, specific references should be added.

•Add the company, city, and country of the used bacterial media and reagents that were used in the biochemical identification of isolates. Also, enumerate all used biochemical reactions.

- Antimicrobial susceptibility testing:

•Illustrate the antimicrobial classes of the tested antimicrobial agents within the text.

•The authors are advised to classify the tested isolates to MDR , XDR, and PDR as described by Magiorakos et al.

Magiorakos AP, Srinivasan A, Carey RB, Carmeli Y, Falagas ME, Giske CG, et al. Multidrug-resistant, extensively drug-resistant and pandrug-resistant bacteria: An international expert proposal for interim standard definitions for acquired resistance. Clin Microbiol Infect. 2012; 18:268–81. doi:10.1111/j.1469-0691.2011.03570.x.

- Why did you ignore the detection of antibiotic resistance genes in the recovered isolates??

•Please use PCR to detect antibiotic resistance genes, followed by gene sequencing if possible. Afterward, the correlation between phenotypic and genotypic multidrug resistance should be performed.

-Add more details about the software used in the statistical analyses.

-Results:

-Add this subtitle: Phenotypic characteristics of the recovered isolates.

•Illustrate in detail the phenotypic characteristics of the recovered isolates.

-Antimicrobial susceptibility testing:

•-Illustrate in a new table the occurrence of MDR (Multidrug resistance) among the recovered isolates (illustrate the names of the antimicrobial classes and different antibiotics):

No. of strains%Type of resistance

R, MDR, and XDRPhenotypic multidrug resistance

(Antimicrobial classes and different antibiotics).The antibiotic -resistance genes

- Increase the resolution of all figures (it should be 600 dpi).

-Discussion:

- The authors are advised to illustrate the real impact of their findings without repetition of results.

-Illustrate the different mechanisms of antimicrobial resistance in S.aureus.

-Conclusion

- Should be rephrased to be sounded. A real conclusion should focus on the question or claim you articulated in your study, which resolution has been the main objective of your paper?

Reviewer #2: Comments to authors:

- The current study has a significant impact, but it needs a major revision:

- The manuscript should be revised for grammar mistakes.

- Please write the scientific names of bacterial pathogens and genes in the correct form all over the manuscript and in the References section (should be italic).

-The title is broad, please modify the title.

- Add more details about the used methods and most prevalent results in the abstract.

-In the introduction: discuss the public health importance of MRSA and its virulence determinants.

-Improve the aim of work.

Methods:

-Explain the methods of isolation and identification in detail??

-Specific references should be added to all the used methods and techniques.

- Antimicrobial susceptibility testing: Add the manufacturing company, city, and country for the used reagents and antimicrobial discs.

-PCR based detection of virulence genes and antimicrobial resistance genes in the most prevalent retrieved bacterial species should be carried out if applicable (or addresses this point in the study limitations)

--Results:

- Discuss in detail the phenotypic characters of the recovered isolates.

-increase the resolution of different Figures: Please improve.

-PCR based detection of virulence genes and antimicrobial resistance genes in the most prevalent retrieved bacterial species should be carried out if applicable (or addresses this point in the study limitations)

-The correlation between the phenotypic and genotypic MDR should be performed.

-Discussion:

- Please improve.

-Please improve the main conclusion of the manuscript.

6. PLOS authors have the option to publish the peer review history of their article (what does this mean?). If published, this will include your full peer review and any attached files.

Reviewer #1: No

Reviewer #2: No

---

## [Author Response · Author response to Decision Letter 0]

18 Feb 2022

Responses to the reviewers comments

Dear Sir/Madam thank you for this careful review of our manuscript. We have responded to each question and comment below : 

(NB : Line number refers to data from manuscript with track changes)

Academic Editor

2- Please provide an amended statement that declares *all* the funding or sources of support (whether external or internal to your organization) received during this study, as detailed online in our guide for authors at http://journals.plos.org/plosone/s/submit-now. 

- Financial disclosure statement has been incorporated in the meta data as suggested and in manuscript financial disclosure statement has also been incorporated.

2- Please also include the statement “There was no additional external funding received for this study.”

- Financial disclosure statement has been incorporated in the meta data as suggested and in manuscript financial disclosure statement has also been incorporated.

4- Please include captions for your Supporting Information files at the end of your manuscript, and update any in-text citations to match accordingly.

- Thank you for indicating, we have included supporting Information file information at Line 166, 239 and 747.

Reviewers 1

Title

I think the work would benefit from the title that contains the main conclusion of the study (should be derived from the conclusion). Please modify the title.

- Title has been modified according to suggestion as follows: 

«High prevalence of Panton-Valentine leukocidin positive, multidrug resistant, Methicillin-resistant Staphylococcus aureus strains circulating among clinical setups in Adamaoua and Far North regions of Cameroon»

Abstract:

- The abstract must illustrate the used methods and the most prevalent results (give more hints about methods and results). Besides, rephrase the aim of the work and the main conclusion of your findings.

More details were included in the abstract highlighting the methods in Line 36 and major results stated in Line 37-58. The conclusion has also been rephrased, Line 59.

Introduction (it needs to be more informative). Give a hint about the virulence factors, different infections caused by MRSA, and the mechanism of disease occurrence.

Response : Further details have been incorporated in introduction Line 79 to 105

The authors should illustrate the public health importance concerning the emergence of multidrug-resistant (MDR) bacterial pathogens that reflect the necessity of new potent and safe antimicrobial agents.

Details are included in Line 118-123 

Rephrase the aim of the work to be clear and better sound.

- The aim of the study has been rephrased. Lines 123-133

Material and methods: Illustrate your methods with subtitles:

Add this subtitle: Bacterial Isolation and identification :

Subtitle has been added at Line 147

Discuss in detail the methods of isolation and identification of S. aureus and MRSA. Besides, specific references should be added.

Details of methods for isolation and identification of S.aureus are provided on line 152 – 165. Moreover suggested references relevant to study are included, reference number 34, 11 and 35. 

Add the company, city, and country of the used bacterial media and reagents that were used in the biochemical identification of isolates. Also, enumerate all used biochemical reactions.

Details of reagents including company and country have been incorporated and given in parenthesis after each Line. Biochemical tests used for preliminary bacterial identification are mentioned in line 155-165.

Antimicrobial susceptibility testing:

Illustrate the antimicrobial classes of the tested antimicrobial agents within the text

Name of each class of antibiotics tested was added and antibiotics were grouped per class

Line 175 - 185

The authors are advised to classify the tested isolates to MDR , XDR, and PDR as described by Magiorakos et al.

Isolates have been classified as suggested and mentioned in line 323 – 325 and presented in Table 6.

Please use PCR to detect antibiotic resistance genes, followed by gene sequencing if possible. Afterward, the correlation between phenotypic and genotypic multidrug resistance should be performed

Resistance gene mec A and virulence factors Luk S/F pvl, sccmec were screening by PCR in this study. It is certainly interesting to perform these additional analysis. However, due to funding limitations selected analysis could be performed. Moreover lab closure as a result of COVID 19 positive status of majority individuals (including 2 of our authors) rendered it impossible to perform additional screenings. Therefore this has been listed as a study limitation in discussion Line 470-473.

Add more details about the software used in the statistical analyses. 

Software used in the statistical analyses are mentioned in line 245-247

Results:

-Add this subtitle: Phenotypic characteristics of the recovered isolates.

The subtitle was added line 254 and more details were provided for the MDR from line 263.

Illustrate in a new table the occurrence of MDR (Multidrug resistance) among the recovered isolates (illustrate the names of the antimicrobial classes and different antibiotics): No. of strains%Type of resistance, MDR, and XDR Phenotypic multidrug resistance (Antimicrobial classes and different antibiotics).The antibiotic -resistance genes

Specific details have been incorporated in new tables 5 and 6 with classes of antibiotics line 322-327 page 12-13

Increase the resolution of all figures (it should be 600 dpi)

Figure 1 has been reformatted to improve resolution of image.

Discussion:

- The authors are advised to illustrate the real impact of their findings without repetition of results.

The discussion part has been restructured to avoid repetition of results.

-Illustrate the different mechanisms of antimicrobial resistance in S.aureus.

Appropriate response is provided in line 414-445, 442-449

Conclusion

Should be rephrased to be sounded. A real conclusion should focus on the question or claim you articulated in your study, which resolution has been the main objective of your paper?

The conclusion has been revised, as suggested Line 476 - 489.

RESPONSE TO THE COMMENTS OF REVIEWER 2

Thank you so much sir/madam for your comments and appreciation to our manuscript. We are going to give responses to your comments in the following paragraphs.

- The manuscript should be revised for grammar mistakes.

Response : The manuscript has been thoroughly revised to remove grammatical mistakes.

- Please write the scientific names of bacterial pathogens and genes in the correct form all over the manuscript and in the References section (should be italic).

Response : Well received, correction have been made throughout the manuscript.

-The title is broad, please modify the title.

Response : New title has been proposed.

- Add more details about the used methods and most prevalent results in the abstract.

Response : Details have been included in the abstract.

-In the introduction: discuss the public health importance of MRSA and its virulence determinants.

Response : Details have been included in introduction Line 79 to 105

-Improve the aim of work.

Response : The aim of the study has been rephrased. Lines 123-133

Methods:

-Explain the methods of isolation and identification in detail??

Response : Details of methods for isolation and identification of S.aureus are provided on line 152 – 165.

-Specific references should be added to all the used methods and techniques.

Response : Thank you for pointing this oversight, we have inserted references in relevant sections. 

- Antimicrobial susceptibility testing: Add the manufacturing company, city, and country for the used reagents and antimicrobial discs.

Response : We have inserted required information in methods section. 

-PCR based detection of virulence genes and antimicrobial resistance genes in the most prevalent retrieved bacterial species should be carried out if applicable (or addresses this point in the study limitations)

Response : Resistance gene mec A and virulence factors Luk S/F pvl, sccmec were screening by PCR in this study. It is certainly interesting to perform these additional analysis. However, due to funding limitations selected analysis could be performed. Moreover lab closure as a result of COVID 19 positive status of majority individuals (including 2 of our authors) rendered it impossible to perform additional screenings. Therefore this has been listed as a study limitation in discussion Line 470-473.

Results:

- Discuss in detail the phenotypic characters of the recovered isolates.

Response : Specific details are provided in the text Line 254-277

-Increase the resolution of different Figures: Please improve.

Response : Figure resolution has been improved.

-PCR based detection of virulence genes and antimicrobial resistance genes in the most prevalent retrieved bacterial species should be carried out if applicable (or addresses this point in the study limitations)

Response :. This has been listed as a study limitation in discussion Line 470-473.

The correlation between the phenotypic and genotypic MDR should be performed.

Response : Phenotypic data is incorporated in new table (table 6) and details are listed in line 322-355.

-Discussion:

- Please improve.

Response : The discussion section has been revised completely.

-Please improve the main conclusion of the manuscript.

Response : The conclusion has been revised.

---

## [Decision Letter · Decision Letter 1]

24 May 2022

High prevalence of Panton-Valentine leukocidin positive, multidrug resistant, Methicillin-resistant *Staphylococcus aureus* strains circulating among clinical setups in Adamawa and Far North regions of Cameroon.

PONE-D-21-37074R1

Dear Dr. Javed,

We’re pleased to inform you that your manuscript has been judged scientifically suitable for publication and will be formally accepted for publication once it meets all outstanding technical requirements.

Kind regards,

Carla Pegoraro

Division Editor

PLOS ONE

Reviewers' comments:

Reviewer's Responses to Questions

**Comments to the Author**

1. If the authors have adequately addressed your comments raised in a previous round of review and you feel that this manuscript is now acceptable for publication, you may indicate that here to bypass the “Comments to the Author” section, enter your conflict of interest statement in the “Confidential to Editor” section, and submit your "Accept" recommendation.

Reviewer #1: (No Response)

Reviewer #2: All comments have been addressed

2. Is the manuscript technically sound, and do the data support the conclusions?

Reviewer #1: Yes

Reviewer #2: Partly

3. Has the statistical analysis been performed appropriately and rigorously? 

Reviewer #1: Yes

Reviewer #2: Yes

4. Have the authors made all data underlying the findings in their manuscript fully available?

Reviewer #1: Yes

Reviewer #2: Yes

5. Is the manuscript presented in an intelligible fashion and written in standard English?

Reviewer #1: Yes

Reviewer #2: Yes

6. Review Comments to the Author

Reviewer #1: The authors have carried out significant changes to the manuscript. They have addressed all the suggested corrections and comments. Really, it's an interesting study that has a significant impact. Now, the manuscript could be accepted.

Congratulations.

Reviewer #2: Most of my previous comments have been addressed. No further comments are needed. Now the manuscript could be accepted.

7. PLOS authors have the option to publish the peer review history of their article (what does this mean?). If published, this will include your full peer review and any attached files.

Reviewer #1: No

Reviewer #2: No

---

## [Editor Report · Acceptance letter]

26 May 2022

PONE-D-21-37074R1 

High prevalence of Panton-Valentine leukocidin positive, multidrug resistant, Methicillin-resistant *Staphylococcus aureus* strains circulating among clinical setups in Adamawa and Far North regions of Cameroon. 

Dear Dr. Javed:

I'm pleased to inform you that your manuscript has been deemed suitable for publication in PLOS ONE. Congratulations! Your manuscript is now with our production department. 

Kind regards, 

on behalf of

Dr Carla Pegoraro 

Staff Editor

PLOS ONE